# Convolutions Die Hard: Open-Vocabulary Segmentation with Single Frozen Convolutional CLIP

**Qihang Yu[1], Ju He[2], Xueqing Deng[1], Xiaohui Shen[1], Liang-Chieh Chen[1]**
[1] ByteDance       [2] The Johns Hopkins University

## Abstract

Open-vocabulary segmentation is a challenging task requiring segmenting and recognizing objects from an open set of categories in diverse environments. One way to address this challenge is to leverage multi-modal models, such as CLIP, to provide image and text features in a shared embedding space, which effectively bridges the gap between closed-vocabulary and open-vocabulary recognition. Hence, existing methods often adopt a two-stage framework to tackle the problem, where the inputs first go through a mask generator and then through the CLIP model along with the predicted masks. This process involves extracting features from raw images multiple times, which can be ineffective and inefficient. By contrast, we propose to build everything into a single-stage framework using a *shared **Frozen Convolutional CLIP*** backbone, which not only significantly simplifies the current two-stage pipeline, but also remarkably yields a better accuracy-cost trade-off. The resulting single-stage system, called FC-CLIP, benefits from the following observations: the *frozen* CLIP backbone maintains the ability of open-vocabulary classification and can also serve as a strong mask generator, and the *convolutional* CLIP generalizes well to a larger input resolution than the one used during contrastive image-text pretraining. Surprisingly, FC-CLIP advances state-of-the-art results on various benchmarks, while running practically fast. Specifically, when training on COCO panoptic data only and testing in a zero-shot manner, FC-CLIP achieve 26.8 PQ, 16.8 AP, and 34.1 mIoU on ADE20K, 18.2 PQ, 27.9 mIoU on Mapillary Vistas, 44.0 PQ, 26.8 AP, 56.2 mIoU on Cityscapes, outperforming the prior art under the same setting by +4.2 PQ, +2.4 AP, +4.2 mIoU on ADE20K, +4.0 PQ on Mapillary Vistas and +20.1 PQ on Cityscapes, respectively. Additionally, the training and testing time of FC-CLIP is $7.5\times$ and $6.6\times$ significantly faster than the same prior art, while using $5.9\times$ fewer total model parameters. Meanwhile, FC-CLIP also sets a new state-of-the-art performance across various open-vocabulary semantic segmentation datasets. Code and models are available at https://github.com/bytedance/fc-clip.

## 1 Introduction

Panoptic segmentation [44] is a complex computer vision task that aims to predict a set of non-overlapping masks, each with its corresponding class label. It combines the tasks of semantic segmentation [37] and instance segmentation [34], making it a challenging problem to solve. Many methods [43, 87, 18, 83, 51, 93, 20, 94, 53] have been proposed to tackle this problem, and a significant progress has been made in terms of panoptic quality (PQ). However, due to the high cost of annotating such a fine-grained dataset [54, 22], the number of semantic classes is typically limited to a few dozens or hundreds. This restriction hinders the further application of existing approaches to real-world settings, where the number of possible semantic classes is unlimited.

37th Conference on Neural Information Processing Systems (NeurIPS 2023).

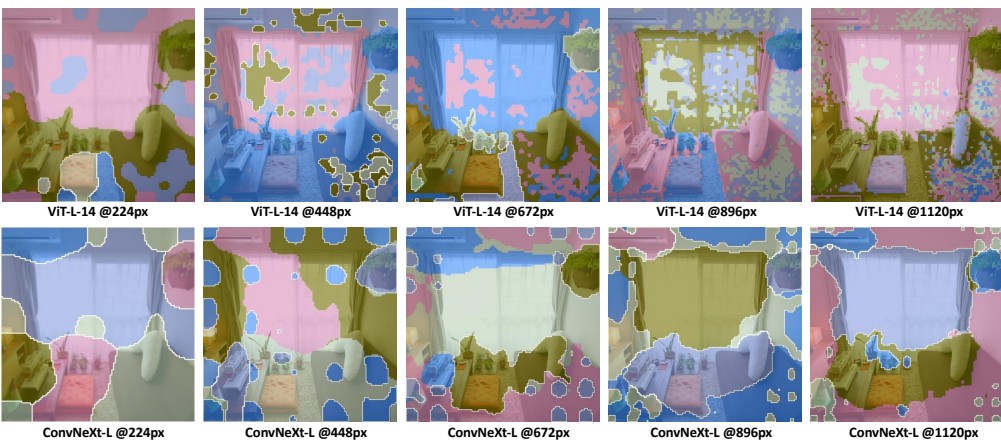

Figure 1: $k$-means visualization on top of frozen CLIP backbone features w.r.t. different input resolutions. Both ViT-based and CNN-based CLIP produces semantic-meaningful features. However, when scaling up the input resolutions, we note that ViT-based CLIP features turn noisier, while CNN-based ones are smoother and generalize better. The smoother feature map is preferable for mask-pooling modules in our design.

To overcome the limitations of closed-vocabulary segmentation, open-vocabulary segmentation [48, 90, 29, 25] has been proposed. These approaches uses text embeddings of category names [97], represented in natural language, as label embeddings, instead of learning them from the training dataset. By doing so, models can classify objects from a wider vocabulary, which improves their ability to handle a broader range of categories. To ensure that meaningful embeddings are provided, a pretrained text encoder [23, 70, 57, 69] is typically used. This encoder can effectively capture the semantic meaning of words and phrases, which is critical for open-vocabulary segmentation.

Multi-modal models, such as CLIP [69] and ALIGN [40], have shown promise for open-vocabulary segmentation due to their ability to learn aligned image-text feature representations from large-scale Internet data [74]. SimBaseline [90] and OVSeg [52] are two recent methods that use a two-stage framework to adapt CLIP for open-vocabulary segmentation. In these methods, images are first processed by a heavy mask generator [36, 20] to obtain mask proposals, and then each masked image crop is generated and fed into a frozen CLIP model for classification. MaskCLIP [25] extends this approach to open-vocabulary panoptic segmentation, but additionally leverages mask proposals as attention masks in the CLIP backbone to efficiently avoid multiple forwarding processes for the masked crops. More recently, ODISE [89] employs a stable diffusion UNet [72, 71] as a frozen backbone for mask generator, which significantly boosts the state-of-the-art performance. However, despite these advances, they still rely on a two-stage framework, where the mask generator and CLIP classifier extract features from raw images separately, resulting in inefficiency and ineffectiveness.

A natural question thus arises as to *whether it is possible to unify the mask generator and CLIP classifier into a single-stage framework for open-vocabulary segmentation*. Sharing the feature extractor between them is a straightforward solution, but it poses two challenges. First, fine-tuning CLIP backbone can disrupt the alignment between image and text features, resulting in a much worse performance on out-of-vocabulary categories. Existing methods [90, 52, 25, 89] rely on another separate backbone for mask generator, increasing model size and computational costs. Second, CLIP models are typically pretrained on relatively lower-resolution inputs, while dense prediction tasks require a much higher resolution for optimal performance. This makes it difficult to directly apply CLIP-pretrained backbones to downstream dense prediction tasks, particularly ViT-based CLIP models [26], where careful treatments are required (*e.g.*, side adapter [17, 91], or cost aggregation [101, 21]). Consequently, existing methods [25, 89] perform mask segmentation and CLIP classification at different input scales, leading to sub-optimal performance.

To alleviate the two challenges, we propose to build both mask generator and CLIP classifier on top of a *shared **F**rozen **C**onvolutional **CLIP*** backbone, resulting in a single-stage framework FC-CLIP. Its

design is based on the following observations. The *frozen* CLIP backbone ensures that the pretrained image-text feature alignment is intact, allowing out-of-vocabulary classification. It can also serve as a strong mask generator by appending a lightweight pixel decoder and mask decoder [20, 94]. The *convolutional* CLIP, based on a Convolutional Neural Network (CNN) [47], empirically shows a better generalization ability compared to ViT-based CLIP [26], when the input size scales up. This echoes the success of fully convolutional networks [60] in dense prediction tasks. Both observations are critical for developing a single-stage framework, but they have been overlooked and undiscovered by existing two-stage pipelines [25, 89]. In Fig. 1, we visualize the learned visual representation of ViT-based and CNN-based CLIP via $k$-means clustering [59]. As shown in the figure, the features learned by CNN-based CLIP are more robust across different input sizes.

Surprisingly, the adoption of a *single frozen convolutional* CLIP as the shared feature extractor results in an extremely simple yet effective design. Specifically, the single-stage FC-CLIP consists of three modules built upon a shared frozen convolutional CLIP backbone: a class-agnostic mask generator, an in-vocabulary classifier, and an out-of-vocabulary classifier (see Fig. 2 for comparison between pipelines). The proposed method not only enjoys a simple design, but also comes with a very low cost for both training and testing. As a comparison, our model has only 238M frozen parameters and 21M trainable parameters, against the state-of-the-art work ODISE [89] that has 1494M frozen and 28M trainable parameters. Furthermore, our model training only takes 25.6 V100 GPU days, which is $7.5\times$ faster compared to ODISE's 192 V100 GPU days. During inference, our model also runs $6.6\times$ faster. Although FC-CLIP enjoys a simple design, it still outperforms previous methods across multiple datasets. Trained on COCO panoptic dataset only, FC-CLIP surpasses prior state-of-the-art ODISE [89] significantly in a zero-shot manner. Specifically, FC-CLIP achieves 26.8 PQ ($+3.4$), 18.2 PQ ($+4.0$), and 44.0 PQ ($+20.1$) on ADE20K, Mapillary Vistas, and Cityscapes, respectively.

As panoptic segmentation unifies semantic and instance segmentation, FC-CLIP naturally extends to open-vocabulary semantic and instance segmentation. With the same model trained on COCO panoptic data only (*i.e.*, no task-specific fine-tuning), FC-CLIP achieves state-of-the-art performance on open-vocabulary instance and semantic segmentation. Specifically, FC-CLIP achieves 16.8 AP on ADE20K, surpassing the state-of-art ODISE [89] by $+2.4$. FC-CLIP also outperforms the state-of-art specialized open-vocabulary semantic segmentation model SAN [91] by $+1.1$ and $+1.1$ mIoU on the challenging ADE20K-847 (A-847) and PASCAL-Context-459 (PC-459) benchmarks, respectively.

In summary, through the lens of a careful re-design of existing two-stage open-vocabulary segmentation models, we establish a simple, strong, and fast baseline for the community. The proposed FC-CLIP adopts a single-stage framework by exploiting a shared frozen convolutional CLIP, which not only advances the state-of-the-art performances on multiple benchmarks, but also enjoys a practically fast training and inference speed. We hope our study will inspire future research on efficient single-stage open-vocabulary segmentation models.

## 2   Related Work

Vision-language models target at encoding vision and language jointly in a fusion model. Early works [78, 16, 98] extract visual representations by pretrained object detectors and fine-tune on downstream tasks with language supervision. Recently, with the breakthrough of large language models [23, 3], rapid progress has been made in this field. CLIP [69] and ALIGN [40] demonstrate that pretraining dual-encoder models with contrastive objectives on large-scale noisy image-text pairs can learn representation with cross-modal alignment ability and show strong performance in zero-shot downstream tasks. The following works [95, 1, 92] further confirm these points and achieve impressive results in zero-shot transfer learning such as open-vocabulary image recognition.

Closed-vocabulary segmentation can be divided into three types according to the semantics of the grouping pixels, *i.e.* semantic, instance and panoptic segmentation. Semantic segmentation interprets high-level category semantic concepts. Prior works [9, 72, 10, 11, 13, 28, 96, 86, 99, 30] mainly treat this task as a per-pixel classification problem and build their models on top of the idea of FCN [60]. Instance segmentation groups foreground pixels into different object instances. Starting from Mask R-CNN [36], prior works [42, 56, 12, 6, 2, 8, 80, 84, 66] mainly address this task with mask classification, where a set of bounding boxes and binary masks are predicted. Panoptic segmentation seeks for holistic scene understanding including both stuff and things. The pioneering work [44] and prevalent ones [55, 43, 87, 18, 50, 82, 14, 67] decompose the problem into various proxy tasks and merge the

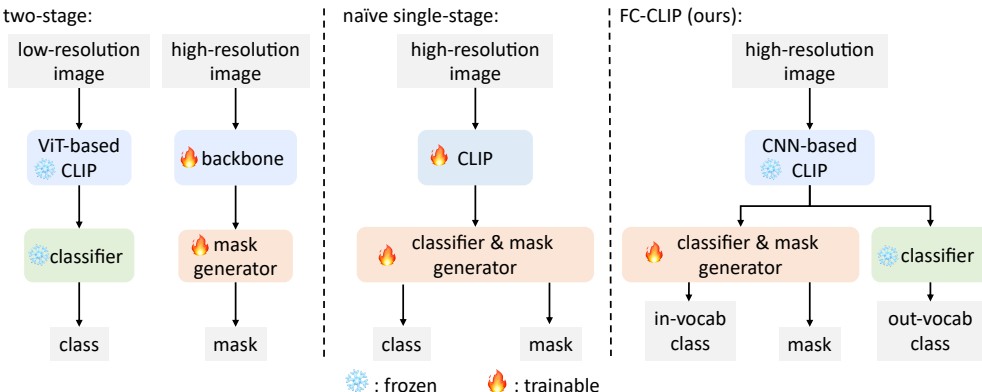

Figure 2: **Comparisons between open-vocabulary panoptic segmentation pipelines.** *Left*: Existing methods [25, 89] adopt a two-stage pipeline, where the first stage employs a high-resolution image to generate class-agnostic masks, and the second stage feeds both the low-resolution image and predicted masks to a frozen CLIP backbone for open-vocabulary recognition. This incurs heavy computation, as image features are extracted multiple times. *Middle*: A naïve single-stage framework builds everything together and fine-tunes the CLIP backbone, breaking the pretrained alignment between images and texts. *Right*: Our single-stage framework FC-CLIP employs a shared frozen convolutional CLIP, where "frozen CLIP" maintains the open-vocabulary recognition and can serve as a strong mask generator, and "convolutional CLIP" generalizes well to large input sizes. Note that the predicted masks are used for CLIP recognition in all three schemes (not shown for simplicity).

results in the end. Recently, following DETR [7], most works [83, 76, 19, 20, 51, 93, 94, 39, 49, 77] present end-to-end solutions based on the idea of mask classification. Standing on their shoulders, our proposed method builds on top of the pixel decoder and mask decoder of Mask2Former [20] by additionally exploiting the open-vocabulary recognition ability from CLIP [69].

Open-vocabulary segmentation aims at segmenting arbitrary classes including those that can not be accessed during the training procedure. Priors works [48, 29, 90, 52, 24, 88, 101, 91, 104, 62, 102, 32] perform open-vocabulary semantic segmentation through leveraging large pretrained vision-language models [69, 40, 71]. Recently, MaskCLIP [25] presents a two-stage pipeline, which consists of a class-agnostic mask generator and a frozen CLIP [69] encoder for cross-modal alignment, and thus expands the scope of the CLIP models into open-vocabulary panoptic segmentation. ODISE [89] digs out the innate potential of pretrained text-image diffusion models [71] in terms of the ability to present open concepts in the representation space for performing strong open-vocabulary panoptic segmentation. FreeSeg [68] encodes multi-granularity concepts into a compact textural abstraction, enabling generalizability to arbitrary text description. Unlike those methods, we propose a single-stage framework by exploiting a single frozen convolutional CLIP backbone, resulting in a simpler, faster, and stronger model than existing works.

We also note that the pioneering work F-VLM [46] builds an open-vocabulary detection framework on top of a frozen CLIP backbone. However, FC-CLIP differs from it with a totally different observation and motivation. Specifically, our work was initially motivated by the state-of-art open-vocabulary segmentation model ODISE [89], which found that the CLIP backbone extracts noisier features than diffusion models (Figure B. 1. in [89]), leading to inferior segmentation results (which justifies their adoption of diffusion models). Their observation motivated us to look deeply into the problem. Interestingly, our discoveries show that both ViT-based (used by ODISE [89]) and CNN-based CLIP can produce semantic-meaningful features. However, when scaling up the input resolutions, we discover that ViT-based CLIP features turn noisier, while CNN-based ones are smoother and generalize better across input sizes. F-VLM [46] also empirically found that a frozen CLIP can provide meaningful features for object detection. However, they did not choose CNN-based CLIP on purpose and thus did not compare carefully between ViT-based and CNN-based CLIP backbones. On the other hand, in our paper, we have provided careful ablation studies on ViT-based and CNN-based CLIP, where we observe that even though both ViT-based and CNN-based CLIP initially have comparable performance at resolution 224, CNN-based CLIP shows better and more robust performance when input resolution scales up.

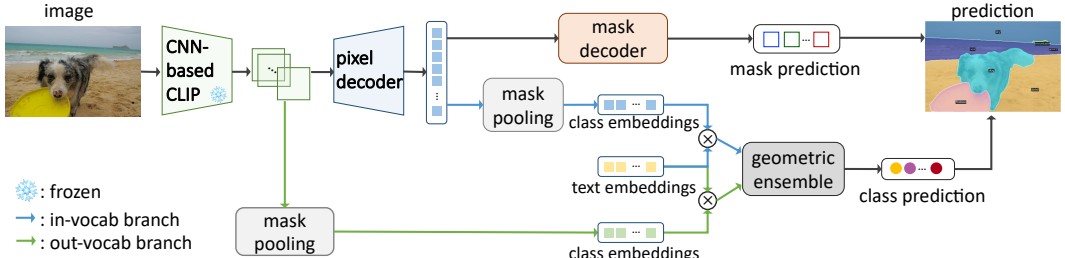

Figure 3: **Overview of FC-CLIP,** which contains three main components: mask generator, an in-vocabulary (in-vocab) classifier, and an out-of-vocabulary (out-vocab) classifier. All components build on top of a shared *frozen covolutional* CLIP backbone. The pixel decoder and mask decoder follow the design of Mask2Former, and generate class-agnostic masks. The in-vocabulary classifier yields the class embeddings by mask-pooling over final pixel features from pixel decoder. During testing, FC-CLIP additionally exploits the out-of-vocabulary classifier by mask-pooling over frozen CLIP backbone features, and the final class prediction is obtained by geometric ensembling both classifiers. Note that the text embeddings are obtained by feeding category names into a CLIP text encoder, which are done beforehand and cached in memory, thus causing no additional costs. Also, the class-agnostic mask proposals are fed to the mask pooling modules (not shown for simplicity).

## 3   Method

In this section, we first define the problem of open-vocabulary segmentation. We then introduce the existing two-stage pipeline, followed by our proposed single-stage framework FC-CLIP.

**Problem Definition**   Open-vocabulary segmentation aims to segment the image $\mathbf{I} \in \mathbb{R}^{H \times W \times 3}$ into a set of masks with associated semantic labels:

$$\{y_i\}_{i=1}^{K} = \{(m_i, c_i)\}_{i=1}^{K}. \tag{1}$$

The $K$ ground truth masks $m_i \in \{0,1\}^{H \times W}$ contain the corresponding ground truth class label $c_i$. During training, a fixed set of class labels $C_{train}$ is used, while during inference, another set of categories $C_{test}$ is used. In the open-vocabulary setting, $C_{test}$ may contain novel categories unseen during training, *i.e.*, $C_{train} \neq C_{test}$. We follow previous works [25, 89] and assume the availability of the category names of $C_{test}$ (represented in natural language) during testing.

**Two-Stage Open-Vocabulary Segmentation**   Existing works [90, 52, 25, 89] adopt a two-stage pipeline for open-vocabulary segmentation. The first stage contains a class-agnostic mask generator $\mathcal{M}$ with parameters $\theta_{\mathcal{M}}$ that generates a set of $N$ mask proposals $\{\hat{m}_i\}_{i=1}^{N} \in \mathbb{R}^{N \times H \times W}$, given the input image $\mathbf{I}$:

$$\{\hat{m}_i\}_{i=1}^{N} = \mathcal{M}(\mathbf{I}; \theta_{\mathcal{M}}). \tag{2}$$

In the second stage, a CLIP adapter $\mathcal{P}$ takes both image $\mathbf{I}$ and mask proposals $\{\hat{m}_i\}_{i=1}^{N}$ as inputs, where the latter input is used to guide the frozen CLIP model $CLIP^*$ (* denotes frozen). The adapter performs mask classification through forwarding processes with either masked crops [90, 52] or masked attention [25, 89]:

$$\{\hat{c}_i\}_{i=1}^{N} = \mathcal{P}(\mathbf{I}, \{\hat{m}_i\}_{i=1}^{N}; CLIP^*), \tag{3}$$

where $\{\hat{c}_i\}_{i=1}^{N} \in \mathbb{R}^{N \times |C|}$ refers to the predicted class probabilities for the $N$ predicted masks, $C \in \{C_{train}, C_{test}\}$ depending on training or testing phase, and $|C|$ is the category size.

Although this framework has achieved impressive open-vocabulary segmentation performance, it has two limitations. First, the image features are extracted *twice*, once for mask generation and the other for mask classification. The double feature extractions incur heavy computation, making it costly to scale up backbone parameters. Second, the mask generator often requires high-resolution inputs (*e.g.*, $1024 \times 1024$), whereas the CLIP model is usually pretrained with lower-resolution images (*e.g.*,

$224 \times 224$). The two-stage pipeline thus needs to feed high-resolution images into the mask generator and low-resolution images into the CLIP classifier, making the model inefficient.

**Naïve Single-Stage Open-Vocabulary Segmentation** To avoid increasing the model size and computational cost of duplicate feature extractions, one may naïvely formulate everything together into a single-stage framework $\mathcal{F}$, where both mask generator and mask classifier share the same CLIP-pretrained backbone $CLIP$ (not frozen) for extracting features from an input image $\mathbf{I}$:

$$\{\hat{m}_i, \hat{c}_i\}_{i=1}^N = \mathcal{F}(\mathbf{I}; CLIP, \theta_M).\tag{4}$$

However, we empirically discover that fine-tuning this naïve single-stage framework causes a misalignment between image and text features in the pretrained CLIP model, leading to sub-optimal performance, especially for novel unseen classes. It also increases the training costs by $2.1\times$ to 52.8 GPU days. Interestingly, our experiments also show that a frozen CLIP backbone can provide sufficient features for mask generation, while preserving the image-text aligned representation. Nevertheless, we still face another challenge, where CLIP models are usually pretrained on low-resolution images (*e.g.*, $224 \times 224$), whereas segmentation models prefer higher-resolution inputs (*e.g.*, $800 \times 1333$ for COCO, or $1024 \times 2048$ for Cityscapes). This discrepancy results in the significant performance degradation, when applying a frozen CLIP on large input images. Digging into the details, we found that it is related to the popular ViT [26] backbone used in CLIP that does not transfer well to different input sizes, which could be alleviated by extra careful designs (*e.g.*, side adapter [17, 91], or cost aggregation [101, 21]). On the other hand, CNN-based CLIP models (such as ResNet [35] and ConvNeXt [58]) exhibit better generalization ability to different input sizes, due to their fully convolutional nature [60]. Additionally, the CNN-based CLIP backbone, extracting multi-scale feature maps, can be used as a simple plug-in module into modern closed-vocabulary segmentation models [20, 94]. Motivated by the observations, we thus propose FC-CLIP, a simple yet effective single-stage open-vocabulary segmentation framework built entirely on a *single frozen convolutional* CLIP backbone $CLIP_{CNN}^*$:

$$\{\hat{m}_i, \hat{c}_i\}_{i=1}^N = \mathcal{F}(\mathbf{I}; CLIP_{CNN}^*, \theta_M).\tag{5}$$

**FC-CLIP** The proposed FC-CLIP leverages the semantic features of a frozen CNN-based CLIP backbone for both mask generation and CLIP classification. Unlike previous works [90, 52, 25, 89], which often train a separate mask generator and ignore the potential reuse of CLIP's semantic features, we incorporate the CNN-based CLIP backbone into the state-of-the-art segmentation method Mask2Former [20]. We note that FC-CLIP is a general meta-architecture that can build on top of several modern segmentation methods [20, 94]. Our approach offers several advantages. By freezing and sharing the backbone features, our model is significantly more efficient during both training and testing (*i.e.*, avoiding feature duplication). The CNN-based CLIP backbone not only transfers well to different input resolutions (from its pretrained image size), but also generates multi-scale feature maps, seamlessly compatible with modern segmentation methods [20, 94]. At a high level, FC-CLIP consists of three components: class-agnostic mask generator, in-vocabulary classifier, and out-of-vocabulary classifier. We detail each component below.

**Class-Agnostic Mask Generator** Following Mask2Former [20], we use a pixel decoder enhanced with multi-scale deformable attention [103] to improve the features extracted from the frozen CNN-based CLIP backbone. The enhanced pixel features, together with a set of object queries [7, 83], are then passed through a series of mask decoders, where each consists of masked cross-attention [20], self-attention [81], and a feed-forward network. The resulting segmentation logits are obtained by performing a matrix multiplication between the object query and pixel features. The predicted masks are matched with ground-truth masks in a one-to-one manner through Hungarian matching [45] and are supervised accordingly. Moreover, as the number of object queries is often greater than the number of labeled masks, only a subset of predicted masks are optimized through this matching process. We apply no penalty to the remaining unmatched proposals, which ensures that more mask proposals are obtained.

**In-Vocabulary Classifier** Once the mask proposals are predicted, they are classified with category text embedding in a contrastive manner, where the class embeddings for each mask and category text embeddings are projected into a common embedding space. That is, the predicted class probability by in-vocabulary classifier is defined as follows: $\forall i = 1, \ldots, N$

$$\hat{c}_{i,in} = softmax(\frac{1}{T}\left[cos(\mathbf{v}_i, \mathbf{t}_1),\ cos(\mathbf{v}_i, \mathbf{t}_2),\ \cdots,\ cos(\mathbf{v}_i, \mathbf{t}_{|C|})\right]),\tag{6}$$

where $T$ is a learnable temperature parameter with initialization of $0.07$ to control the sharpness of the distribution, $cos$ is cosine distance measurement, $\mathbf{v}_i$ is the class embeddings for $i$-th predicted mask, which is obtained by mask pooling over the *final pixel features from pixel decoder*, similar to [29]. $\mathbf{t}_j$ is the category name's text embeddings of class $j$, which is obtained by feeding the category name to a CLIP-pretrained text encoder. Note that these category text embeddings only need to be generated once. They are then kept in memory to serve as text classifiers, and thus it incurs negligible additional cost during training. This forms our in-vocabulary classifier.

**Out-of-Vocabulary Classifier** During inference, however, we notice that using the in-vocabulary classifier alone fails to generalize to completely novel unseen classes, as the model is only trained on a finite set of categories and thus could not recognize diverse novel concepts. To address this issue, we introduce an out-of-vocabulary classifier, which applies mask pooling to the *frozen CLIP backbone features*, aiming to borrow the pretrained (intact) open-vocabulary recognition ability from CLIP. Unlike the other two-stage methods [90, 52, 25, 89], where one or multiple forward processes of CLIP are needed, the adopted out-of-vocabulary classifier introduces marginal additional costs, since the backbone features are already extracted (and only lightweight mask-pooling is performed). The predicted class probability by out-of-vocabulary classifier $\hat{c}_{i,out}$ is then obtained in a manner similar to Eq. (6) by replacing $\mathbf{v}_i$ with the mask-pooled features over *frozen CLIP backbone features*. This classifier strictly maintains the original CLIP feature distribution, allowing us to better recognize brand new categories. Note that the out-of-vocabulary classifier is only performed during testing.

**Combining In- and Out-of-Vocabulary Classifiers** Following prior works [31, 29, 46, 89], we employ geometric ensemble to fuse the classification scores between in-vocabulary and out-of-vocabulary classifiers. That is, $\forall j = 1, \ldots, |C|$

$$\hat{c}_i(j) = \begin{cases} (\hat{c}_{i,in}(j))^{(1-\alpha)} \cdot (\hat{c}_{i,out}(j))^{\alpha}, & \text{if } j \in C_{train} \\ (\hat{c}_{i,in}(j))^{(1-\beta)} \cdot (\hat{c}_{i,out}(j))^{\beta}, & \text{otherwise} \end{cases} \tag{7}$$

where $\hat{c}_i(j)$ denotes the $j$-th element of $\hat{c}_i$, and the underscripts $in$ and $out$ refer to in-vocabulary and out-of-vocabulary classifier, respectively. $\alpha, \beta \in [0, 1]$ balance the predictions between in- and out-of-vocabulary classifiers for seen and novel unseen categories.

# 4 Experimental Results

Herein, we provide implementation details of FC-CLIP in Sec. 4.1. After setting the stage, we introduce our main results, compared with state-of-the-art methods and ablations studies in Sec. 4.2.

## 4.1 Implementation Details

**Architecture** We use ConvNeXt-Large CLIP [58, 69] backbones from OpenCLIP [38][1] pretrained on LAION-2B [74] dataset. On top of the CLIP backbone, we build the mask generator, following Mask2Former [20]. Nine mask decoders are employed to generate the class-agnostic masks by taking as inputs the enhanced pixel features and a set of object queries. For in-vocabulary classification, following [29], the class embeddings are obtained by mask-pooling the pixel features from the pixel decoder's final output. Afterwards, the classification logits (before softmax) is obtained by matrix multiplication between the predicted class embeddings and categories' text embeddings.

**Training Strategy** We follow [20] and adopt the same training recipe and losses without any special design. The training is optimized with AdamW [41, 61] optimizer and weight decay $0.05$. We use a crop size of $1024 \times 1024$. We employ the learning rate $1 \times 10^{-4}$ and a multi-step decay schedule. The training batch size is 16, and the model is trained for 50 epochs on COCO panoptic training set [54].

**Inference Strategy** During inference, the shorted side of input images will be resized to $800$ while ensuring longer side not exceeds $1333$. For Cityscapes and Mapillary Vistas, we increase the shorter side size to $1024$. We adopt mask-wise merging scheme [20] for the mask predictions. The out-of-vocabulary classifier is only performed during inference by mask-pooling over the frozen CLIP backbone features. The final classification results are then obtained by geometric ensembling in- and out-of-vocabulary classifiers [31, 29, 46, 89], as in Eq. (7), where we default $\alpha = 0.4$ and

---

[1] https://github.com/mlfoundations/open_clip

Table 1: **Open-vocabulary panoptic segmentation performance on ADE20K.** The proposed FC-CLIP demonstrates better performances than prior arts, while using much fewer frozen parameters. We provide more results in the supplementary material

| | params (M) | | zero-shot test dataset ADE20K | | | training dataset COCO | | |
|---|---|---|---|---|---|---|---|---|
| method | frozen | trainable | PQ | AP | mIoU | PQ | AP | mIoU |
| MaskCLIP [25] | 304 | 63 | 15.1 | 6.0 | 23.7 | - | - | - |
| FreeSeg [68] | - | - | 16.3 | 6.5 | 24.6 | - | - | - |
| ODISE [89] | 1494 | 28 | 22.6 | 14.4 | 29.9 | 55.4 | 46.0 | 65.2 |
| ODISE [89] (caption) | 1494 | 28 | 23.4 | 13.9 | 28.7 | 45.6 | 38.4 | 52.4 |
| FC-CLIP (ours) | 200 | 21 | 26.8 | 16.8 | 34.1 | 54.4 | 44.6 | 63.7 |

Table 2: **Open-vocabulary panoptic segmentation performance on street-view datasets**. The proposed FC-CLIP demonstrates better transferability to street-view dataset

| | zero-shot test dataset | | | | | | | | |
|---|---|---|---|---|---|---|---|---|---|
| | Mapillary Vistas | | | | Cityscapes | | | | |
| method | PQ | SQ | RQ | mIoU | PQ | SQ | RQ | AP | mIoU |
| ODISE [89] | 14.2 | 61.0 | 17.2 | - | 23.9 | 75.3 | 29.0 | - | - |
| FC-CLIP (ours) | 18.2 | 57.7 | 22.9 | 27.9 | 44.0 | 75.4 | 53.6 | 26.8 | 56.2 |

$\beta = 0.8$. Following prior arts, we also adopt prompt engineering from [29, 89] and prompt templates from [31, 52]. If not specified, FC-CLIP is only trained on COCO panoptic dataset [54]. Following prior works [29, 89], we zero-shot evaluate the model on ADE20K [100], Cityscapes [22], and Mapillary Vistas [64] for open-vocabulary panoptic segmentation. We also report open-vocabulary semantic segmentation results on those datasets along with PASCAL datasets [27, 63]. The panoptic segmentation results are evaluated with the panoptic quality (PQ) [44], Average Precision (AP), and mean intersection-over-union (mIoU), and semantic segmentation is evaluated with mIoU [27]. Note that all results are obtained with the same single checkpoint trained on COCO panoptic data only.

## 4.2 Results

We summarize the main results for open-vocabulary panoptic segmentation and semantic segmentation in Tab. 1, Tab. 2 and Tab. 3, where we train FC-CLIP on COCO *train* set with panoptic annotation and evaluate it on various datasets in a zero-shot manner.

**Open-Vocabulary Panoptic Segmentation Evaluation on ADE20K**    In Tab. 1, we compare our FC-CLIP with other state-of-the-art methods on ADE20K [100], the main test-bed of zero-shot open-vocabulary panoptic segmentation. As shown in the table, our method achieves significantly better performance compared to MaskCLIP [25], with +11.7 PQ, +10.8 AP and +10.4 mIoU, even though we use fewer frozen (−66M) and trainable (−42M) parameters. When compared to the concurrent methods FreeSeg [68] and ODISE [89], the advantage of FC-CLIP persists. FC-CLIP is +10.5 PQ, +10.3 AP, and +9.5 mIoU better than FreeSeg without using COCO-Stuff annotations [5] (which contains more semantic classes than COCO-Panoptic). Our PQ, AP, mIoU score are also +4.2, +2.4, +4.2 higher than ODISE under the same training settings. Compared to ODISE with caption [15] for supervision, our model still outperforms it by +3.4 PQ, setting a new state-of-the-art record. Meanwhile, it is noticeable that our model has $6.3\times$ ($5.9\times$) significantly fewer frozen (total) parameters compared to ODISE, which utilizes a strong large backbone from stable diffusion [71] for feature extraction.

**Open-Vocabulary Panoptic Segmentation Evaluation on Street-View Datasets**    In Tab. 2, we evaluate on Cityscapes and Mapillary Vistas, which focus on street driving scenes. Compared to state-of-the-art method ODISE, FC-CLIP achieves better performances on both datasets. Specifically, it outperforms ODISE by +4.0 PQ and +20.1 PQ on Mapillary Vistas and Cityscapes, respectively. Notably, FC-CLIP has a slightly lower SQ, which indicates our mask generator is actually weaker than the one in ODISE, which utilizes a much larger backbone.

**Open-Vocabulary Semantic Segmentation Evaluation**    Although our model was trained on COCO panoptic data only, it also performs well on open-vocabulary semantic segmentation. In Tab. 3, we

Table 3: **Open-vocabulary semantic segmentation performance.** The proposed FC-CLIP also demonstrates state-of-the-art performances on open-vocabulary semantic segmentation

| method | training dataset | mIoU | | | | | |
| --- | --- | --- | --- | --- | --- | --- | --- |
| | | A-847 | PC-459 | A-150 | PC-59 | PAS-21 | PAS-20 |
| SPNet [85] | Pascal VOC [27] | - | - | - | 24.3 | 18.3 | - |
| ZS3Net [4] | Pascal VOC [27] | - | - | - | 19.4 | 38.3 | - |
| LSeg [48] | Pascal VOC [27] | - | - | - | - | 47.4 | - |
| GroupViT [88] | GCC [75]+YFCC [79] | 4.3 | 4.9 | 10.6 | 25.9 | 50.7 | 52.3 |
| SimBaseline [90] | COCO Stuff [5] | - | - | 15.3 | - | 74.5 | - |
| ZegFormer [24] | COCO Stuff [5] | - | - | 16.4 | - | 73.3 | - |
| LSeg+ [48, 29] | COCO Stuff [5] | 3.8 | 7.8 | 18.0 | 46.5 | - | - |
| OVSeg [52] | COCO Stuff [5] | 9.0 | 12.4 | 29.6 | 55.7 | - | 94.5 |
| SAN [91] | COCO Stuff [5] | 13.7 | 17.1 | 33.3 | 60.2 | - | 95.5 |
| OpenSeg [29] | COCO Panoptic + COCO Caption | 6.3 | 9.0 | 21.1 | 42.1 | - | - |
| ODISE [89] (caption) | COCO Panoptic + COCO Caption | 11.0 | 13.8 | 28.7 | 55.3 | 82.7 | - |
| MaskCLIP [25] | COCO Panoptic | 8.2 | 10.0 | 23.7 | 45.9 | - | - |
| ODISE [89] | COCO Panoptic | 11.1 | 14.5 | 29.9 | 57.3 | 84.6 | - |
| FC-CLIP (ours) | COCO Panoptic | 14.8 | 18.2 | 34.1 | 58.4 | 81.8 | 95.4 |

Table 4: **FPS comparison.** All results are obtained with one V100 GPU, CUDA 11.6 and PyTorch 1.13, by taking the average runtime on the entire validation set, including post-processing time

| method | ADE20K | COCO |
| --- | --- | --- |
| ODISE [89] | 0.41 | 0.39 |
| FC-CLIP (ours) | 2.71 (6.61$\times$) | 2.76 (7.08$\times$) |

report our model's performance on various benchmarks against other open-vocabulary segmentation models, where FC-CLIP shows an overall superior performance. Specifically, with the same training annotations used, FC-CLIP outperforms MaskCLIP by +6.6, +8.2, +10.4, +12.5 mIoU across A-847, PC-459, A-150, and PC-59, respectively. Compared to methods with caption annotations, FC-CLIP persists its advantages, where it outperforms ODISE (caption) by +3.8, +4.4, +5.4, +3.1 mIoU across datasets A-847, PC-459, A-150, PC-59 respectively. Against other open-vocabulary semantic segmentation methods, our model maintains its advantages across different datasets, despite being trained solely with panoptic annotations. Furthermore, it demonstrates comparable performance to state-of-the-art open-vocabulary semantic segmentation methods, which utilize the COCO-Stuff dataset as their training set. The COCO-Stuff dataset comprises 171 classes, 38 more classes than COCO-Panoptic, and offers highly desirable annotations for semantic segmentation tasks. It is worth mentioning that these methods build their approach on top of ViT-L (with extra designs [91]), resulting in a significantly larger model size compared to our deployed ConvNeXt-L (304M *vs.* 198M). Despite the disparity in model size, FC-CLIP remains competitive in terms of performance. Specifially, FC-CLIP outperforms state-of-the-art open-vocabulary semantic segmentation method SAN [91] by 1.1 and 1.1 mIoU on the challenging A-847 and PC-459 datasets.

**Inference Speed** We provide a comparison of FPS (frames per second) in Tab. 4. The proposed FC-CLIP not only demonstrates superior performances, but also enjoys a significant fast inference time: FC-CLIP runs 6.61$\times$ and 7.08$\times$ faster than ODISE evaluated on ADE20K and COCO datasets, respectively.

**Training on ADE20K and Evaluating on COCO** We further validate the effectiveness of FC-CLIP by using a different training dataset. Specifically, we follow [68, 89] to train our model on ADE20K dataset with panoptic annotation, and evaluate it on COCO panoptic dataset. As shown in Tab. 5, FC-CLIP outperforms FreeSeg [68] by +10.5 PQ, and ODISE [89] by +2.0 PQ on COCO dataset. Notably, our model actually has a lower SQ ($-1.4$) compared to ODISE, which utilizes a much larger backbone and thus has a stronger mask generator. Nevertheless, FC-CLIP still outperforms ODISE significantly with a simple yet effective design.

**Fine-tuning CLIP Backbone Harms Performance on Novel Vocabularies** We validate the necessity of freezing CLIP backbone to ensure a better generalization to novel vocabularies. We compare the performance of trainable CLIP variant and frozen CLIP variant in Fig. 4, where we use the same mask proposals to ensure a fair comparison. Specifically, we compare the performance on

Table 5: **Results of training on ADE20K panoptic and evaluating on COCO panoptic val set.** The proposed FC-CLIP performs better than prior arts, even in the different setting (*i.e.*, trained on ADE20K and zero-shot evaluated on COCO)

| | zero-shot test dataset | | | training dataset | | |
| | COCO | | | ADE20K | | |
| method | PQ | SQ | RQ | PQ | SQ | RQ |
|---|---|---|---|---|---|---|
| FreeSeg [68] | 16.5 | 72.0 | 21.6 | - | - | - |
| ODISE [89] | 25.0 | 79.4 | 30.4 | 31.4 | 77.9 | 36.9 |
| FC-CLIP (ours) | 27.0 | 78.0 | 32.9 | 41.9 | 78.2 | 50.2 |

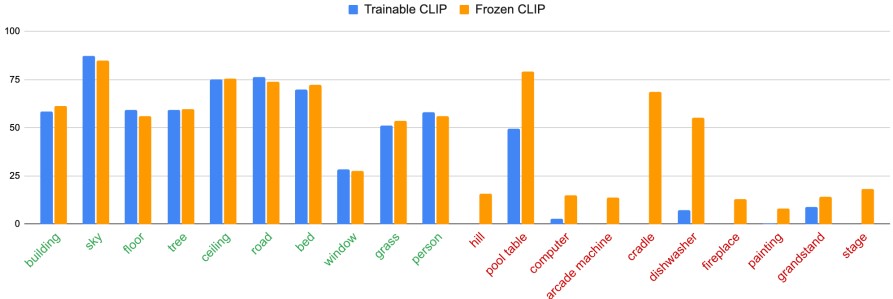

Figure 4: **Trainable CLIP *vs*. Frozen CLIP, with per-class PQ analysis.** We show 10 common classes (labeled in green) shared by COCO and ADE20K, and 10 novel classes (labeled in red) that are only in ADE20K. The frozen CLIP demonstrates a much better recognition ability for novel classes, while performing similarly for the seen classes.

10 seen classes, which are shared by both COCO and ADE20K (*e.g.*, person, sky), and 10 unseen classes, which are only included in ADE20K dataset (*e.g.*, arcade machine, dishwasher). As shown in the figure, tuning CLIP backbone leads to a worse performance on unseen concepts, which breaks the CLIP feature alignment and thus loses its recognition ability on a much wider vocabulary.

## 5   Conclusion

In this work, we have presented FC-CLIP, a simple yet effective single-stage framework for open-vocabulary segmentation. FC-CLIP shows great potential by building everything on top of a shared frozen convolutional CLIP backbone, which not only significantly reduces training and testing costs, but also establishes a strong baseline on multiple benchmarks. Our study demonstrates how to better adapt a pretrained CLIP model for downstream dense prediction tasks, which we hope will shed the light on unleashing CLIP's potential for other various downstream tasks.

**Limitations**   FC-CLIP presents a simple single-stage open-vocabulary segmentation framework with state-of-the-art performance. We note that there exist some interesting research topics to be explored in the near future, such as better unleashing CLIP's potential in both mask segmentation and classification, how to deal with conflict or overlapping vocabularies (*e.g.*, cat *vs*. cat head), *etc*.

**Broader Impact**   FC-CLIP shows great potential for segmenting and naming every object in the scene, which could facilitate many applications including intelligent home assistants, robots, self-driving, *etc*. Yet it relies on CLIP model pre-trained on the Internet data that may be biased, which calls for future research for calibration to avoid misuse.

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

**Appendix** In the following supplementary materials, we present additional experimental results pertaining to the design of FC-CLIP. Our supplementary analysis also includes comparisons against other methods that specifically address open-vocabulary semantic segmentation, ensemble methods, and hyperparameter tuning. Furthermore, we provide a quantitative comparison between ViT-based CLIP and CNN-based CLIP across varying input sizes, along with additional visualizations and comprehensive dataset details.

# 6 Additional Experimental Results

**Fine-tuning or Freezing CLIP Backbone in FC-CLIP** In this study, we provide a comprehensive analysis of the impact of fine-tuning or freezing the CLIP backbone in our framework. We specifically focus on the $PQ^{seen}$ and $PQ^{unseen}$ metrics, which evaluate the performance for classes that overlap and do not overlap between the training and testing datasets, respectively. To determine whether a class is seen or unseen, we adopt the prompt engineering technique described in [29], which provides synonyms or subcategories of classes. Specifically, if any category name in test dataset overlaps with a category name in training dataset, we consider it as a seen class; otherwise unseen. As discussed in the main paper, the proposed FC-CLIP contains three components: a class-agnostic mask generator, an in-vocabulary classifier, and an out-of-vocabulary classifier. We thus explore using frozen or trainable CLIP for each component, and summarize the results in Tab. 6. To ensure a fair comparison, all "trainable" modules utilize the same weights, resulting in identical mask proposals and in-vocabulary classification results. Moreover, we note that the first row in Tab. 6 with trainable mask generator and in-vocabulary classifier, can be considered as an approximation to OpenSeg [29] in our framework. Our findings reveal that an in-vocabulary classifier built upon a trainable CLIP backbone achieves a higher $PQ^{seen}$ score (37.9 compared to 32.4), but experiences a decrease in $PQ^{unseen}$ (2.6 compared to 12.6) compared to a frozen out-of-vocabulary classifier. Consequently, a model that incorporates a trainable CLIP backbone for all components yields a PQ of 24.1, which is 2.7 lower than our final model (last row) that relies on a single frozen CLIP backbone. Using a trainable mask generator and in-vocabulary classifier, along with a frozen out-of-vocabulary classifier boosts the performance but requires maintaining one trainable and one frozen CLIP weights, resulting in $2\times$ more backbone parameters. In summary, our observations demonstrate that building the entire framework upon a frozen CLIP backbone is not only effective but also efficient, providing a better balance between $PQ^{seen}$ and $PQ^{unseen}$ metrics.

Table 6: **Effects of fine-tuning or freezing the CLIP backbone for each module in FC-CLIP.** Building all three modules upon a single frozen CLIP backbone attains best performance. Note that our mask generator and in-vocabulary classifier use the same backbone following [20, 29, 94], and thus it is infeasible (denoted as N/A) for the setting in the 2nd last row. Our final setting is labeled in gray

| mask generator | in-vocabulary classifier | out-of-vocabulary classifier | PQ | $PQ^{seen}$ | $PQ^{unseen}$ |
|---|---|---|---|---|---|
| trainable | trainable | - | 17.7 | 37.9 | 2.6 |
| trainable | - | frozen | 21.1 | 32.4 | 12.6 |
| trainable | trainable | trainable | 24.1 | 38.9 | 13.1 |
| trainable | trainable | frozen | 25.4 | 40.0 | 14.6 |
| trainable | frozen | frozen | N/A | N/A | N/A |
| frozen | frozen | frozen | 26.8 | 39.5 | 17.3 |

**Evaluation with Grounding PQ and Grounding mIoU** It is worth emphasizing that despite the absence of grounding loss [33, 97, 29, 89] during training, our model exhibits exceptional grounding segmentation capabilities. Tab. 7 presents the grounding PQ and grounding mIoU scores of FC-CLIP, following the evaluation methodology outlined in [29]. In this evaluation, we exclusively employ ground-truth classes as text query inputs to assess the effectiveness of concept grounding. Compared to OpenSeg [29], FC-CLIP achieves a substantial performance improvement, with notable enhancements of $+11.6$, $+9.1$, $+13.1$, and $+17.7$ on A-847, PC-459, A-150, and PC-59, respectively. Even when compared to OpenSeg trained with the Localized Narrative dataset [65], which enables training on a significantly larger vocabulary, FC-CLIP still surpasses it with improvements of $+8.0$, $+2.2$, $+8.6$ and $+13.4$ on A-847, PC-459, A-150 and PC-59, respectively, underscoring the grounding proficiency of FC-CLIP.

Table 7: **Grounding segmentation performance.** The proposed FC-CLIP also demonstrates state-of-the-art performances on grounding segmentation. MV: Mapillary Vistas

| method | grounding PQ | | | grounding mIoU | | | | | |
|---|---|---|---|---|---|---|---|---|---|
| | ADE20K | Cityscapes | MV | A-847 | PC-459 | A-150 | PC-59 | PAS-21 | PAS-20 |
| ALIGN [40, 29] | - | - | - | 17.8 | 21.8 | 25.7 | 34.2 | - | - |
| ALIGN w/ proposal [40, 29] | - | - | - | 17.3 | 19.7 | 25.3 | 32.0 | - | - |
| LSeg+ [48, 29] | - | - | - | 10.5 | 17.1 | 30.8 | 56.7 | - | - |
| OpenSeg [29] | - | - | - | 21.8 | 32.1 | 41.0 | 57.2 | - | - |
| OpenSeg [29] w/ L. Narr | - | - | - | 25.4 | 39.0 | 45.5 | 61.5 | - | - |
| FC-CLIP (ours) | 38.4 | 48.1 | 21.5 | 33.4 | 41.2 | 54.1 | 74.9 | 88.7 | 98.5 |

Table 8: **Ensemble methods comparison with zero-shot evaluation (PQ) on ADE20K.** Our method is robust to different ensemble methods (arithmetic and geometric). The results show that it is preferable to bias towards using the in-vocabulary classifier for seen classes and the out-of-vocabulary classifier for unseen classes. Our final setting ($\alpha = 0.4, \beta = 0.8$) is labeled in gray

| method | arithmetic | geometric |
|---|---|---|
| ($\alpha = 0.0, \beta = 0.0$) | 17.8 | 17.8 |
| ($\alpha = 1.0, \beta = 1.0$) | 21.9 | 21.9 |
| ($\alpha = 0.0, \beta = 1.0$) | 25.3 | 25.3 |
| ($\alpha = 1.0, \beta = 0.0$) | 17.5 | 17.5 |
| ($\alpha = 0.5, \beta = 0.5$) | 25.0 | 25.3 |
| ($\alpha = 0.5, \beta = 0.6$) | 25.6 | 26.4 |
| ($\alpha = 0.5, \beta = 0.7$) | 25.5 | 26.7 |
| ($\alpha = 0.5, \beta = 0.8$) | 25.4 | 26.6 |
| ($\alpha = 0.4, \beta = 0.6$) | 25.1 | 25.6 |
| ($\alpha = 0.4, \beta = 0.7$) | 25.6 | 26.4 |
| ($\alpha = 0.4, \beta = 0.8$) | 25.6 | 26.8 |
| ($\alpha = 0.4, \beta = 0.9$) | 25.4 | 25.8 |

Table 9: **Quantitative results of ViT-based CLIP and CNN-based CLIP when input size (denoted as "res") varies for panoptic segmentation on COCO and ADE20K.** All results are obtained by applying CLIP directly as a mask classifier with the same mask proposals from ODISE [89]

| CLIP backbone | COCO PQ @res | | | | | ADE20K PQ @res | | | | |
|---|---|---|---|---|---|---|---|---|---|---|
| | 224 | 448 | 672 | 896 | 1120 | 224 | 448 | 672 | 896 | 1120 |
| ViT-L/14 | 19.3 | 22.5 | 20.6 | 18.5 | 14.9 | 11.9 | 13.7 | 12.6 | 11.6 | 9.1 |
| ConvNeXt-L | 17.3 | 23.5 | 27.0 | 28.6 | 29.3 | 9.3 | 12.8 | 14.8 | 16.0 | 15.9 |

**Ensemble In-Vocabulary and Out-of-Vocabulary Classifiers**    In Tab. 8, we present experiments conducted to evaluate the impact of ensemble methods and ensemble parameters on the performance of the in-vocabulary and out-of-vocabulary classifiers. Specifically, we examine two ensemble methods: arithmetic and geometric. The arithmetic method involves a linear combination of the in-vocabulary classifier and the out-of-vocabulary classifier, while the geometric method is defined as shown in Equation (7) of main paper. It is worth noting that FC-CLIP exhibits robustness to different ensemble methods, with both methods displaying a consistent trend within the explored hyper-parameter ranges. However, the geometric ensemble consistently outperforms the arithmetic ensemble by a slight margin. Additionally, we observe that preference is given to values of $\alpha \leq 0.5$ and $\beta \geq 0.5$, which biases the model towards using the in-vocabulary classifier for seen classes and the out-of-vocabulary classifier for unseen classes. We also explore extreme cases, including $\alpha = 0.0$ and $\beta = 0.0$ (i.e., exclusively utilizing the in-vocabulary classifier for every class), $\alpha = 1.0$ and $\beta = 1.0$ (i.e., exclusively utilizing the out-of-vocabulary classifier for every class), $\alpha = 0.0$ and $\beta = 1.0$ (i.e., using the in-vocabulary classifier for seen classes and the out-of-vocabulary classifier for unseen classes), and $\alpha = 1.0$ and $\beta = 0.0$ (i.e., using the out-of-vocabulary classifier for seen classes and the in-vocabulary classifier for unseen classes). The results align with our observations that it is preferable to bias towards the in-vocabulary classifier for seen classes and the out-of-vocabulary classifier for unseen classes.

Table 10: **Quantitative results of ViT-based CLIP and CNN-based CLIP when input size (denoted as "res") varies for ImageNet-1k classification.**

| CLIP backbone | Accuracy @res | | | | | | |
|---|---|---|---|---|---|---|---|
| | 224 | 336 | 448 | 560 | 672 | 784 | 896 |
| ViT-L/14 | 75.3 | 74.3 | 71.3 | 67.5 | 63.1 | 58.5 | 53.9 |
| ConvNeXt-L | 75.1 | 77.1 | 76.8 | 74.2 | 69.8 | 65.6 | 58.4 |

Table 11: **Open-vocabulary segmentation performance with different backbones and segmentation frameworks.** All models are trained on COCO and tested on the other datasets in a zero-shot manner. MV: Mapillary Vistas. ∗: kMaX-DeepLab with multi-scale deformable attention [103]

| method | backbone | panoptic datasets (PQ) | | | semantic datasets (mIoU) | | | |
|---|---|---|---|---|---|---|---|---|
| | | ADE | Cityscapes | MV | A-847 | PC-459 | PC-59 | PAS-21 |
| FC-CLIP | R50 [35, 69] | 17.9 | 40.3 | 15.9 | 7.1 | 12.9 | 50.5 | 75.9 |
| FC-CLIP | R101 [35, 69] | 19.1 | 40.9 | 16.7 | 7.7 | 12.3 | 48.9 | 77.6 |
| FC-CLIP | R50×4 [69] | 21.8 | 42.2 | 17.4 | 8.7 | 13.1 | 54.0 | 79.0 |
| FC-CLIP | R50×16 [69] | 22.5 | 42.0 | 17.8 | 10.3 | 15.7 | 56.4 | 80.7 |
| FC-CLIP | R50×64 [69] | 22.8 | 42.7 | 18.2 | 10.8 | 16.2 | 55.7 | 80.3 |
| FC-CLIP w/ kMaX | ConvNeXt-L [58, 38] | 24.5 | 43.0 | 17.0 | 11.4 | 15.0 | 57.4 | 84.7 |
| FC-CLIP w/ kMaX∗ | ConvNeXt-L [58, 38] | 26.4 | 40.2 | 17.4 | 13.6 | 17.5 | 57.1 | 81.2 |
| FC-CLIP | ConvNeXt-L [58, 38] | 26.8 | 44.0 | 18.2 | 14.8 | 18.2 | 58.4 | 81.8 |

**Quantitative ViT-based CLIP *vs*. CNN-based CLIP when Input Size Scales**  Training our model solely with ViT-based CLIP, without any additional modifications [101, 17, 91, 21], is infeasible. Furthermore, applying ViT to large input sizes is computationally expensive. Therefore, to evaluate the effects of using ViT- or CNN-based CLIP in our framework, we incorporate them into our out-of-vocabulary classifier, which is performed only during inference. To ensure a fair comparison, we use the same mask proposals and disable the geometric ensemble scheme. We also perform experiment on the ImageNet [73] benchmark to ensure a comprehensive comaprison. In Tab. 9 and Tab. 10, we conduct an ablation study to analyze the impact of different input resolutions for CLIP models. We consider both ViT-based (ViT-L/14) and CNN-based (ConvNeXt-L) CLIP models. By employing them as zero-shot classifiers and varying the input resolutions, we observe that CNN-based CLIP demonstrates superior generalization ability as the input size scales up. Specifically, we observe that the ViT-L/14 CLIP has a higher PQ and Accuracy at a lower resolution (*i.e.*, input size 224), but suffers from a higher resolution, which leads existing two-stage methods [90, 52, 25, 91, 89] to adopt different input resolutions for mask generator and classifier branches. On the contrary, FC-CLIP provides a simple solution by adopting a CNN-based CLIP that generalizes well to different input sizes.

**FC-CLIP with Different Backbones and Different Segmentation Frameworks**  Though we majorly report FC-CLIP results with ConvNeXt-L [58, 69] backbone in Mask2Former [20] framework. We note that FC-CLIP can be easily incorporated with different backbones and segmentation frameworks. Specifically, we experiment FC-CLIP with different backbones (*e.g*., ResNet [35]) and different segmentation architecture (*e.g*., kMaX-DeepLab [94]). As shown in Tab. 11, FC-CLIP demonstrates superior performance across different backbones and frameworks.

**Visualization**  We provide visualization on ADE20K *val* set in Fig. 5.

# 7   Datasets Information and Licenses

The datasets we used for training and/or testing FC-CLIP are described as follows.

**COCO:**  We train FC-CLIP on COCO data with panoptic annotation [54]. We follow the 2017 splits which include $118k$ images for *train* split and $5k$ images for *val* split. If not specified, we train our model on the COCO *train* split and report results on *val* set of various datasets.

License: Creative Commons Attribution 4.0 License

URL: https://cocodataset.org/#home

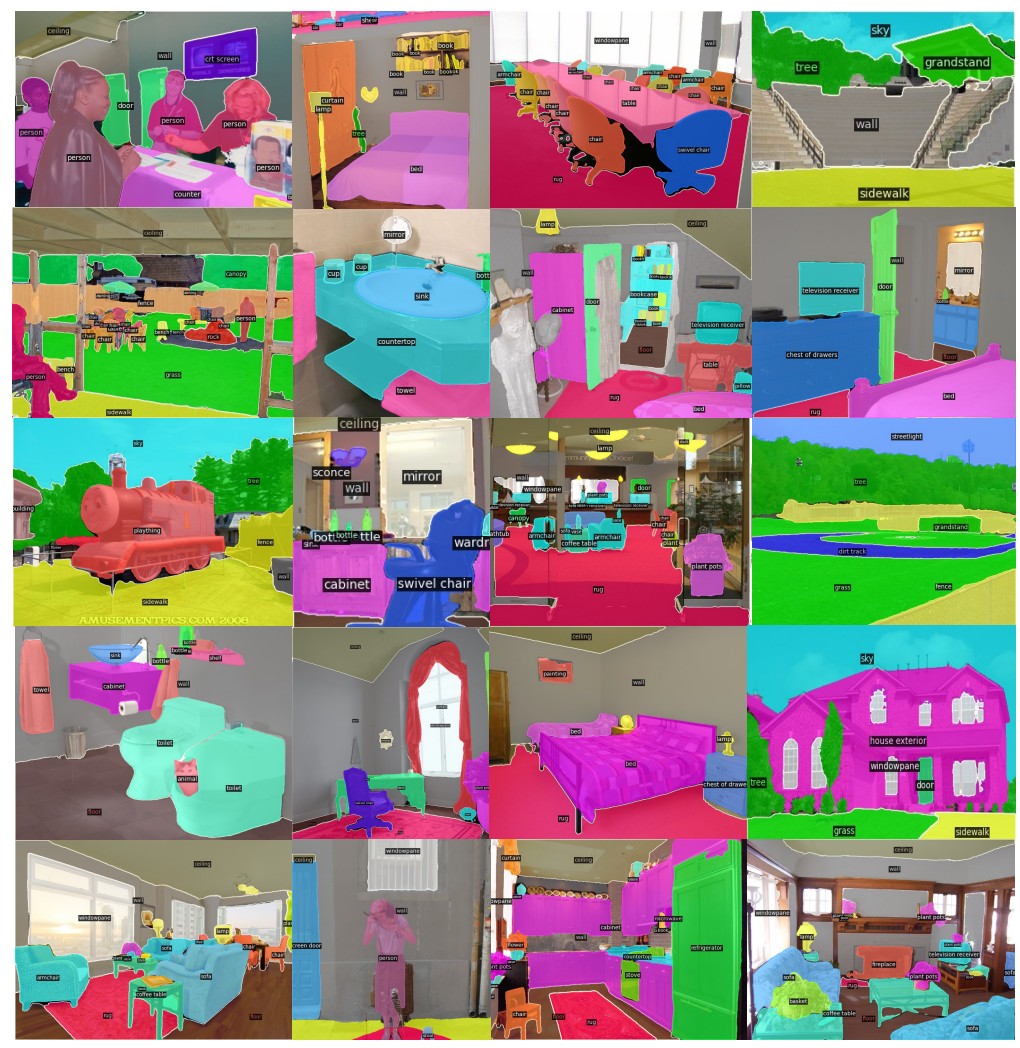

Figure 5: **Visualization examples of FC-CLIP on ADE20K *val* set.** FC-CLIP is trained on COCO panoptic training set and zero-shot evaluated on ADE20K validation set.

**ADE20k:** ADE20k [100] covers a wide range of indoor and outdoor scenes, with $2k$ *val* images. We evaluate FC-CLIP on both the version with 847 classes (A-847) and the more widely-used version with 150 frequent categories (A-150).

License: Creative Commons BSD-3 License

URL: `https://groups.csail.mit.edu/vision/datasets/ADE20K/`

**Cityscapes:** Cityscapes [22] focuses on semantic understanding of urban street scenes. We use the *fine* data includes 500 images for validation set.

License: This dataset is made freely available to academic and non-academic entities for non-commercial purposes such as academic research, teaching, scientific publications, or personal experimentation.

URL: `https://www.cityscapes-dataset.com/`

**Mapillary Vistas:** Mapillary Vistas [64] is a large-scale traffic-related dataset, including $2k$ images for validation purposes.

License: Creative Commons Attribution NonCommercial Share Alike (CC BY-NC-SA) license

URL: `https://www.mapillary.com/dataset/vistas`

**Pascal Context:** Pascal Context [63] covers a wide variety of indoor and outdoor scenes and includes $5k$ *val* images. We evaluate FC-CLIP on both its full version (PC-459) with 459 classes and the more common version (PC-59) with 59 classes.

URL: `https://www.cs.stanford.edu/~roozbeh/pascal-context/`

**Pascal VOC:** Pascal VOC [27] contains $1.5k$ *val* images with 20 foreground classes and 1 background class. Due to the ambiguity in definition of "background", we assign the background class to the pixels predicted as PC-59 categories that are not in Pascal VOC following [29], which leads to PAS-21. We also evaluate the model with background class excluded, which leads to PAS-20.

URL: `http://host.robots.ox.ac.uk/pascal/VOC/`

