# OpenReview forum: "Convolutions Die Hard: Open-Vocabulary Segmentation with Single Frozen Convolutional CLIP"
_NeurIPS.cc/2023/Conference — NeurIPS 2023 poster_

### Official Review · Reviewer_VW2t · 2023-06-13

**Soundness:** 3 good
**Presentation:** 4 excellent
**Contribution:** 3 good
**Rating:** 6
**Confidence:** 4

**Summary:**

The paper tackles the open-vocabulary panoptic segmentation with a frozen CLIP. Unlike previous two-stages works, the paper uses the same backbone from the frozen CLIP for the mask generator and classifier. The single-stage framework accelerates the training and inferring speed and achieves SOTA performance.

**Strengths:**

The single-stage framework proposed in the paper is concise and time-friendly for training and inference. The writing of the paper is clear and can be easily followed. The paper shows the convolutional CLIP has a better generalization ability to higher-resolution inputs than ViTs. FC-CLIP achieves SOTA performance on multiple open-vocabulary panoptic segmentation benchmarks.

**Weaknesses:**

1. The novelty of the paper is weak. The idea of using the frozen CLIP for open-vocabulary prediction is the same as F-vlm. The main difference is that paper uses the framework of F-vlm to tackle the panoptic segmentation problem and substitute the Mask R-CNN heads in F-vlm with kMaX-DeepLab. Regarding innovation, I am not saying that the author must come up with some fancy innovations. But compared with the predecessors' work, what targeted improvements and discoveries have been made to new problems? Compared to F-vlm, I did not see any special design for the panoptic segmentation problem.

2. The claim that (frozen) CNN-based CLIP is better than ViT-based CLIP for dense prediction is not well verified. In the paper, the author verified the claim in Figure 1 with k-means visualization of these two types of CLIP. But whether the smoother feature correlates with the dense prediction performance is unclear. Many works [2][3] use ViT-based CLIP as the backbone of dense prediction tasks. Although the author compares ViT-based CLIP and CNN-based CLIP in Table 5 of Supp, the mask proposals are optimized with the CNN feature (on seen classes). It is more rigorous to use the ground truth mask to evaluate the classification performance of these two types of CLIP. The experiment of training a mask proposal on top of ViT-based CLIP is also needed to evaluate the mask proposal performance of ViT-based CLIP. The results of the 1281 resolution (used in the main paper) are missing in Table 5 of Supp.

3. Eq. 7 might have some typos. 'i' is the index of the mask, and 'i' should be substituted with the index of the class. Eq. 7 might be copied from F-vlm because in F-vlm,  'i' denotes the index of the class, which is correct. The author should check the formula more carefully.


[1] Weicheng Kuo, Yin Cui, Xiuye Gu, AJ Piergiovanni, and Anelia Angelova. F-vlm: Open-vocabulary object detection upon frozen vision and language models. ICLR, 2023.

[2] Ma, Chao et al. “Open-vocabulary Semantic Segmentation with Frozen Vision-Language Models.” British Machine Vision Conference (2022).

[3] Zhou, Ziqi et al. “ZegCLIP: Towards Adapting CLIP for Zero-shot Semantic Segmentation.” ArXiv abs/2212.03588 (2022): n. pag.

**Questions:**

1. In Open-CLIP, they have three configures of ConvNext-Large:

ConvNext-Large (D) @ 256x256 /w augreg on LAION-2B with a top-1 of 75.9%

ConvNext-Large (D) @ 320x320 fine-tune of 256x256 weights above for ~2.5B more samples on LAION-2B, top-1 of 76.6%

ConvNext-Large (D) @ 320x320 soup of 3 fine-tunes of 256x256 weights above on LAION-2B, top-1 of 76.9%

Which one did the author use? The ConvNext-L trained with 320x320 is naturally more suitable for high-resolution inputs than ViT-L/14 trained with 224x224.

**Limitations:**

In limitations, the author mentioned "how to deal with conflict or overlapping vocabularies (e.g. cat vs. cat head)". Does this problem occur in the experiment of the paper? What will happen to FC-CLIP if it faces the problem?

---

> ### Author Rebuttal · Authors · 2023-08-09
>
> We sincerely thank the reviewer for the constructive feedback and address the concerns below.
>
> > ***W1: Novelty, comparison to F-VLM***
>
> Please refer to **C1: Relationship to F-VLM** and **C2: Novelty/Contribution**
>
> > ***W2: ViT-based or CNN-based CLIP for dense prediction***
>
> We thank the reviewer for the questions, and address the concerns carefully below.
>
> **Many works [2][3] use ViT-based CLIP**: We thank the reviewer for the extra references, which we will add in the revision. We note that those papers [2][3] adopting ViT-based CLIP work *specifically* for semantic segmentation only, which often does not require a very high input resolution as it does not need to predict object-level masks with different scales (COCO dataset particularly contains many small objects). To be concrete, a typical ADE20K semantic segmentation setting is to use $512 \times 512$ input resolution, while for COCO panoptic segmentation, it is usually $800 \times 1333$ or $1281 \times 1281$. The ViT-L/14 (with output stride 14) is sufficient under the relatively lower resolution $512 \times 512$. Additionally, we note that state-of-the-art open-vocabulary semantic segmentation method SAN [24] also observes that ViT performs undesirable under higher resolution and thus they have to adopt a two-stage framework to feed different resolution images to the model. We quote from their paper: *"Accurate semantic segmentation needs high-resolution images, but the released ViT CLIP models are designed for low-resolution images (e.g., $224 \times 224$) and directly apply to high-resolution images giving a poor performance. To alleviate the conflicts in input resolutions, we use low resolution images in the CLIP model and high-resolution images in the side adapter network."*
>
> **Table 5 of Supp, the mask proposals are optimized with the CNN feature; mask proposals from ViT-based CLIP**: Regarding the evaluation of directly applying CNN-based and ViT-based CLIP as mask classifier, we use the **ODISE model as mask generator**, which is **not specifically trained for either CNN-based or ViT-based CLIP (but used as a separate module to ensure fairness)**, and thus the mask proposals were not optimized with CNN-based CLIP. Finally, we would like to note that false positive/negative proposals are a common issue in most segmentation models, and thus evaluating the CLIP model with a real mask proposal model (instead of ground-truth masks) is also practically important and reasonable. We will revise the draft to make it clearer.
>
> **Results of 1281 resolution are missing in Table 5 of Supp**: We thank the reviewer for the suggestion. We note that either 1280 or 1281 is not divisible by 14, and thus it is not applicable to use ViT-L/14. We provide more results under resolution 224 to 1120 below, where the 1120 is expected to provide an approximation result of 1280.
>
> |            |      | COCO |  PQ  | @res |      |    |      | ADE20K |  PQ  | @res |      |
> |------------|:----:|:----:|:----:|:----:|:----:|----|:----:|:------:|:----:|:----:|:----:|
> |            |  224 |  448 |  672 |  896 | 1120 | \| |  224 |   448  |  672 |  896 | 1120 |
> | ViT-L/14   | 19.3 | 22.5 | 20.6 | 18.5 | 14.9 | \| | 11.9 |  13.7  | 12.6 | 11.6 |  9.1 |
> | ConvneXt-L | 17.3 | 23.5 | 27.0 | 28.6 | 29.3 | \| |  9.3 |  12.8  | 14.8 | 16.0 | 15.9 |
>
> > ***W3: Equation typos***
>
> We sincerely thank the reviewer for pointing out the typos. We will fix them in the revision. Additionally, we would like to note that the geometric ensemble is not a specific design proposed by F-VLM. Instead, it is a common trick employed in many prior works such as [SAN, OVSeg, SimSeg, ViLD, ODISE].
>
> > ***Q1: ConvNeXt-L CLIP backbone comparison***
>
> We thank the reviewer for the question. We use the ConvNeXt-L with "laion2b_s29b_b131k_ft_soup" weight in OpenCLIP. However, we respectfully disagree that ConvNeXt-L enjoys an advantage by pre-training with a higher resolution. In fact, as we work on object-level prediction, the dense feature map is required, instead of a globally pooled vector. Additionally, we note that only the last layer of the CLIP model can be used directly for classification due to the pre-training objective. Consequently, the ConvNeXt-L backbone leads to a ***32x*** downsampled feature map (i.e., $10 \times 10$, if input size is $320 \times 320$), while ViT-L/14 has a much larger feature map (i.e., $16 \times 16$, even if input size is $224 \times 224$), which actually favors ViT-L/14 when using them as dense feature extractors. This explains why ViT-L/14 performs better than CovnNeXt-L when input size is $224 \times 224$ in Table 5 of Supplementary.
>
> > ***Limitation: Overlapping vocabularies***
>
> We thank the reviewer for the question. We note that this is a common problem for most existing open-vocabulary segmentation works. It also exists in current benchmarks, e.g., in ADE20K, there are three different but semantically similar classes: *chair*, *armchair*, and *swivel chair*. FC-CLIP simply relies on CLIP to make predictions among the overlapping vocabularies. We agree with the reviewer that how to resolve such problems (e.g., building hierarchical vocabulary space) is an interesting future direction.

---

### Official Review · Reviewer_5KPy · 2023-07-04

**Soundness:** 3 good
**Presentation:** 3 good
**Contribution:** 3 good
**Rating:** 5
**Confidence:** 4

**Summary:**

This paper proposes to build a one-stage open-vocabulary detector with a frozen language encoder (CLIP) and acvhieves good performance.

**Strengths:**

1. Strong performance. Performance on many benchmarks is better than previous models.
2. Simple framework. One stage does look more simple to use.

**Weaknesses:**

1. The novelty is limited. The proposed design does not involve enough novelty.
2. From my point of view, one stage and two stage methods do not differ that much. Why one-stage method is much better than previous methods?

**Questions:**

Described above.

**Limitations:**

Described above.

---

> ### Author Rebuttal · Authors · 2023-08-09
>
> We sincerely thank the reviewer for the constructive feedback and address the concerns below.
>
> > ***W1: Novelty***
>
> Please refer to common concerns **C2: Novelty/Contribution**
>
> > ***W2: One-stage v.s. two-stage methods***
>
> A one-stage framework can re-use a shared feature extractor across different modules, which not only leads to a simpler framework but also achieves much better accuracy-cost trade-off. Compared to the state-of-the-art two-stage method ODISE, FC-CLIP demonstrates a better performance, while using only 15.4% total parameters and runs 4.4x faster. The proposed one-stage model FC-CLIP significantly simplifies the system design without the need to consider separate modules in the pipeline (e.g., only need to consider one backbone instead of multiple backbone combinations).

---

### Official Review · Reviewer_YCke · 2023-07-08

**Soundness:** 3 good
**Presentation:** 3 good
**Contribution:** 2 fair
**Rating:** 3
**Confidence:** 5

**Summary:**

The authors propose an approach based on CLIP model and extend it for zero-shot semantic segmentation. The authors argue that pervious works solve the problem with a two-stage approach, which first generates mask predictions using one backbone and then extract features from another backbone using CLIP model is suboptimal and inefficient. They present an approach using a frozen CLIP image encoder as backbone and generate mask and prediction with trainable pixel and mask decoders. They validate the effectiveness of proposed method on several commonly used semantic segmentation benchmarks and achieve solid performance.

**Strengths:**

1. Extending CLIP model to semantic segmentation in an efficient and effective approach is a recently popular and active research problem in Computer Vision. The authors present an approach to address this problem.

2. The proposed technique is technically sound.

3. The authors conduct experiments comparing to the latest previous work, ODISE, and shows that the performance is comparable while enjoy faster inference speed.

**Weaknesses:**

1. The contribution is incremental. The proposed approach in very similar to OpenSeg [28], with just different selection of backbone, mask generation and classification of in-vocabulary classification. The minor difference in the selections of components is not significant for the top-tier conference. Comparing to [28], it is unclear where the improvement come from. It could be because of the selection of pixel decoder features for in-vocabulary categories, or the different selection of mask generation or the input resolution. There is no ablation to understand this and neither of these changes are significant contributions.

2. The naive single-stage baseline presented in the paper is not reasonable. As CLIP backbone is fine-tuned without considering text encoder, it naturally loses the capability of open-vocabulary classification. The baseline is not reasonable but created in a way to makes the contributions of proposed method appear meaningful

3. The technical details presented in this paper are not self-contained. Details as follow:
- The description of naive single-stage baseline is vague. What is the mask generator used in the baseline? Does the classifier use text embedding or something else?
- The description of class-agnostic mask generator presented in the paragraph starting at line 192 is vague and lack of details. What is the pixel decoder with axial attention? What is the kMax mask decoders? What is the k-means cross-attention? How is Hungarian matching used? How is the subset of predicted masks selected during the matching process? As mask generation is a critical component in the proposed method, it needs to be properly described even if they are introduced in the previous work to make the paper self-contained and to justify the contribution of the proposed method.

- Abuse of notation. In line 136, sum{m_i} <= 1^{HxW} is not properly defined. I am guessing it means every entry in the matrix is smaller or equal to 1. This is a minor issue.

**Questions:**

1. Please clarify the contributions of proposed methods.

2. Please justify the different between proposed approach and the previous work [28]

**Limitations:**

The limitation is discussed in the paper to some extent and I don't have major concerns on the limitation.

---

> ### Author Rebuttal · Authors · 2023-08-09
>
> We thank the reviewer for the constructive feedback and address the concerns below.
>
> >***W1, Q2: Comparison to OpenSeg***
>
> In the submission, we have already carefully discussed the limitations of naive single-stage methods such as OpenSeg, and how FC-CLIP differs from them. We summarize them again below for reference:
>
> 1. As discussed in the Supplementary L19 to L23, we note that methods like OpenSeg jointly fine-tune the whole model, which leads to a worse generalization ability to novel concepts. While in this paper, we have provided in-depth analysis and experiments to validate the importance of adopting a frozen CNN-based CLIP backbone for better open-vocabulary segmentation performance.
>
> 2. We respectfully disagree that *”Comparing to [28], it is unclear where the improvement come from. …. There is no ablation to understand this and neither of these changes are significant contributions."* In fact, in the Supplementary, we have compared a naive single-stage baseline which can be considered as a reproduction of OpenSeg (as their code is not open-sourced) in our framework (Supp. Table 1). We quote the results from Supplementary here:
>
> | mask generator | in-vocab classifer | out-of-vocab classifier |  $PQ$  | $PQ_{seen}$ | $PQ_{unseen}$ |
> |:--------------:|:------------------:|:-----------------------:|:----:|:-------------:|:---------------:|
> |    trainable   |      trainable     |            -            | 16.2 |      33.9     |       2.9       |
> |    trainable   |          -         |          frozen         | 19.4 |      28.5     |       12.6      |
> |    trainable   |      trainable     |        trainable        | 19.6 |      34.9     |       8.2       |
> |    trainable   |      trainable     |          frozen         | 23.8 |      36.0     |       14.7      |
> |     frozen     |       frozen       |          frozen         | 24.5 |      37.4     |       14.9      |
>
> As shown in the table above, the first row **(trainable, trainable, -)** can be considered as a reproduction of OpenSeg in our framework. Both our final model (last row) and our "OpenSeg" baseline (1st row) use the same backbone and segmentation frameworks thus involve no "selections of components".
>
> Besides, we also disagree that *"improvement is from better selection of modules".* When compared to prior state-of-the-art ODISE, FC-CLIP has a much smaller backbone and simpler design, and can actually have an even better performance when switching to ODISE’s segmentation framework Mask2Former.
>
> >***W2: The naive single-stage baseline not reasonable. Fine-tuning text encoder?***
>
> We respectfully disagree with the reviewer. To the best of our knowledge, most open-vocabulary segmentation methods (e.g., SAN, ODISE) do not fine-tune the text encoder. This is natural, as the COCO Panoptic dataset contains only 133 classes, where freezing the text encoder avoids the catastrophic forgetting problem. Our pipeline closely follows them. We do not see that freezing the text branch is improper, considering segmentation datasets only contain limited training vocabularies. To the best of our knowledge, *only* the concurrent work OpenSeeD (ICCV 2023) fine-tunes the text encoder when trained on COCO and Objects365 datasets. However, they achieve 19.7 PQ on ADE20K, which is similar to the performance of our all trainable baseline 19.6 PQ (only trains on COCO). We also note that the single-stage method OpenSeg is not open-sourced. But, we have closely followed it to build our solid single-stage baseline, which is not *"created in a way to make the contributions of the proposed method appear meaningful".*
>
> >***W3.1: Mask generator and classifier for naive single-stage baseline***
>
> As shown in the Figure 2 of our paper, we aim to provide a system-wise comparison among different open-vocabulary segmentation methods, involving no specific requirements on Mask Generator or Classifier. To answer this question, we have built a baseline mimicking OpenSeg (as their code/model is not open-sourced) in Table 1 of Supplementary, which uses the same mask generator and classifier as our final model (kMaX-DeepLab segmentation framework and CLIP text embeddings).
>
> >***W3.2: Technical details regarding model architecture and training***
>
> We respectfully disagree with the reviewer. The proposed FC-CLIP is a meta architecture, which can be built on top of several segmentation frameworks ***without*** any change (as shown in our response to Q1 of Reviewer N4iz, where we also experiment with Mask2Former). We do not provide any further details of the adopted kMaX-DeepLb segmentation framework, as they are not the main focus of this work (and we did not make any change to them). Providing additional detailed descriptions of them obscures the focus of this work: a general open-vocabulary segmentation pipeline with a ***frozen convolutional CLIP***. We believe that the provided descriptions and references in the draft are already sufficient for the readers to get the context. Additionally, we promise to fully open-source our training/testing codes, allowing the community to check every detail. If the reviewer still disagrees with us, we would like to get the second opinion from either the other reviewers or the area chair. If they also think so, we are more than happy to add more details for those segmentation frameworks in the main paper.
>
> >***W3.3: Notation***
>
> The notation appropriately explains the property of panoptic segmentation, where the predicted masks do not overlap with each other. This notation is standard for panoptic segmentation and is aligned to the panoptic segmentation works [40, 72]. That being said, we thank reviewer for suggestion, and we will make it clear.
>
> >***Q1: Contributions of proposed methods.***
>
> Please refer to common concerns **C2: Novelty/Contribution**

---

### Official Review · Reviewer_GVRs · 2023-07-11

**Soundness:** 3 good
**Presentation:** 4 excellent
**Contribution:** 3 good
**Rating:** 6
**Confidence:** 4

**Summary:**

This work proposes a new approach to open-vocabulary panoptic segmentation that unifies the mask generator and CLIP classifier into a single-stage framework. This is achieved by sharing the feature extractor between them, which presents two challenges: disrupting the alignment between image and text features during fine-tuning and the need for higher resolution inputs for dense prediction tasks. The authors address these challenges by adopting the shared Frozen CNN-based CLIP backbone. The resulting FC-CLIP model achieves state-of-the-art performance on several benchmarks while being more efficient and effective than previous methods.

**Strengths:**

1. The paper's approach to unifying the mask generator and CLIP classifier into a single-stage framework is a novel contribution to the field of open-vocabulary panoptic segmentation.

2. The paper is well-written and clearly presents the problem of open-vocabulary panoptic segmentation, the challenges of the current two-stage pipeline, and the proposed FC-CLIP model.

3. The paper is well-organized and clearly presents the problem, methodology, and results. The authors provide a detailed explanation of the FC-CLIP model, making it easy to understand the proposed approach. The paper's figures and tables are well-designed and provide a clear visualization of the results.

**Weaknesses:**


1. Absence of Ablation Study: The paper lacks an ablation study to conduct a thorough analysis of the impact of different components or design choices of the FC-CLIP model on its performance. For instance, the inclusion of an in/out-vocabulary classifier and the combine strategy, if adopted within the naive single-stage framework, what would be its effect on performance?

2. The experimental results are not convincing enough. The FC-CLIP model has a higher number of trainable parameters compared to ODISE. However, it remains uncertain whether the observed performance improvement can be attributed solely to the increased parameter count. (2) Table 2 highlights significant improvements achieved by the FC-CLIP model on the Cityscapes dataset. It would be beneficial to provide additional explanations or discussions to elucidate the reasons behind these improvements. (3) While FC-CLIP demonstrates noticeable enhancements on the ADE20K dataset, the improvements on the COCO dataset appear to be comparatively smaller.


**Questions:**

The proposed method is well-motivated and novel. However, some key experiments are lack. More details please refer to the weakness part.

**Limitations:**

The authors have briefly discussed the limitations and future work of their work. However, there are some limitations required to discussed.

The FC-CLIP model relies on a CLIP model pre-trained on the Internet data that may be biased, which calls for future research for calibration to avoid misuse.
They could provide a more detailed discussion of these issues. For example, they could discuss the potential biases in the pre-trained CLIP model and how these biases could impact the performance of the FC-CLIP model. They could also discuss the potential negative societal impact of the FC-CLIP model, such as its use in surveillance systems or other applications that could infringe on privacy rights.

---

> ### Author Rebuttal · Authors · 2023-08-09
>
> We sincerely thank the reviewer for the constructive feedback and address the concerns below.
>
> >***W1: Missing ablation study on different components/design choices***
>
> We have **already provided the asked ablation studies** on model designs in the Table 1 and Table 4 of Supplementary, e.g., combining in/out-of-vocabulary classifiers, as well as the effect of them in the naive single-stage framework.
> To clarify the confusion, we quote the results from supplementary here again:
>
> **1. Combining in-/out-of-vocabulary classifiers and hyper-parameters:**
>
> | ensemble (alpha, beta) | arithmetic ensemble | geometric ensemble |
> |:------------------------:|:-------------------:|:------------------:|
> | (0.0, 0.0)             |         17.6        |        17.6        |
> | (1.0, 1.0)             |         19.4        |        19.4        |
> | (0.0, 1.0)             |         22.8        |        22.8        |
> | (1.0, 0.0)             |         15.7        |        15.7        |
> | (0.5, 0.5)             |         23.2        |        23.4        |
> | (0.4, 0.6)             |         23.1        |        23.3        |
> | (0.4, 0.7)             |         23.7        |        24.2        |
> | (0.4, 0.8)             |         24.0        |        24.5        |
> | (0.4, 0.9)             |         23.9        |        24.1        |
>
> **2. Effect of each module:**
>
> | mask generator | in-vocab classifer | out-of-vocab classifier |  $PQ$  | $PQ_{seen}$ | $PQ_{unseen}$ |
> |:--------------:|:------------------:|:-----------------------:|:----:|:-------------:|:---------------:|
> |    trainable   |      trainable     |            -            | 16.2 |      33.9     |       2.9       |
> |    trainable   |          -         |          frozen         | 19.4 |      28.5     |       12.6      |
> |    trainable   |      trainable     |        trainable        | 19.6 |      34.9     |       8.2       |
> |    trainable   |      trainable     |          frozen         | 23.8 |      36.0     |       14.7      |
> |     frozen     |       frozen       |          frozen         | 24.5 |      37.4     |       14.9      |
>
> For the naive single-stage models with in-/out-of-vocabulary classifiers, please refer to the third row **(trainable, trainable, trainable)**, which leads to 19.6 PQ with a performance degradation by -4.9 PQ from our final setting in the last row ($PQ_{unseen}$ degrades most by -6.7 PQ). The 4th row **(trainable, trainable, frozen)** provides the result by employing another frozen off-the-shelf CLIP as out-of-vocabulary classifier and leads to a comparable PQ score 23.8 (-0.7 PQ), but we note that this actually leads to the same model costs as two-stage models, with the need of additional weights and extra forwarding pass. We will add additional elaboration to the results to avoid confusion.
>
> >***W2.1: Improvement comes from more trainable parameters?***
>
> We note that ODISE has a great advantage over our method with *1294M* more frozen parameters. We only have *6M* more trainable parameters, which is mainly due to the differences between kMaX-DeepLab and Mask2Former. We think that the 7.5 times more frozen parameters should have more impacts on performance compared to small additional 6M trainable parameters. That being said, to further address the question, we also build FC-CLIP on top of Mask2Former, which not only has fewer (-7M) trainable parameters (note that ODISE has another trainable MLP for diffusion model's text embeddings), but also has a better performance than ODISE, indicating the improvement comes from our simple and effective meta architecture design.
>
> |                     | frozen params (M) | trainable params (M) |         ADE20K        | Mapillary Vistas | Cityscapes |
> |---------------------|:-----------------:|:--------------------:|:---------------------:|:----------------:|:----------:|
> | ODISE               |        1494       |          28          | 22.6 / 23.4 (caption) |       14.2       |    23.9    |
> | FC-CLIP-kMaX        |        200        |          34          |          24.5         |       17.0       |    43.0    |
> | FC-CLIP-Mask2Former |        200        |          21          |          26.8         |       18.2       |    44.0    |
>
> >***W2.2: Improvement on Cityscapes***
>
> FC-CLIP has a better improvement on the street-view datasets such as Cityscapes and Mapillary Vistas than ODISE. We think that it is because ODISE relies on latent diffusion models for feature extraction, which performs a VQ tokenization on the input image. The VQ tokenizer was trained for image generation purposes and thus may not handle complex street view very well. On the contrary, FC-CLIP has a simpler and more effective design, which generalizes well to such datasets.
>
> >***W2.3: Improvement on COCO***
>
> In Table 1 of the main paper, the COCO dataset is used as the only training dataset for both ODISE and FC-CLIP. Therefore, its results only indicate the closed-vocabulary performance and thus closely related to the model's capacity to fit the training dataset instead of its ability to novel datasets in the open-vocabulary scenario. In this case, ODISE has a much larger model size and thus is expected to fit COCO better than FC-CLIP (as shown in the paper L281 to  L284, we also observe that ODISE can provide better mask proposals, thanks to its much larger model size). However, FC-CLIP generalizes much better to other datasets in a zero-shot manner.
>
> >***CLIP biases and limitations***
>
> We thank the reviewer for bringing up the bias and limitation in the pre-trained CLIP model, which may also impact FC-CLIP, we will add related limitation discussion in a revision per suggested.

---

### Official Review · Reviewer_N4iz · 2023-07-23

**Soundness:** 3 good
**Presentation:** 4 excellent
**Contribution:** 3 good
**Rating:** 7
**Confidence:** 5

**Summary:**

In this submission, the authors proposed a new method for open-vocabulary panoptic segmentation. In the open-vocabulary segmentation setting, the model is trained on seen category annotations and tested on unseen categories. The frozen features of CLIP/ALIGN have been demonstrated to be effective in new category generalization.  To leverage the representation of CLIP, prior works typically use the two-stage framework, one stage forwards high-resolution images for mask generation with vanilla networks, and the other stage inputs low-resolution images for mask classification. The proposed a shared Frozen Convolutional CLIP backbone (FC-CLIP) which unifies the pipelines into a single stage. It exploits frozen CLIP with ConvNeXt image encoder as the backbone network. FC-CLIP is simple and yet effective, surpass prior state-of-the-art on many open-vocabulary segmentation tasks.

**Strengths:**

1. This submission explains the motivation and method very well. I like Figure 2 very much, which clearly shows the differences between FC-CLIP and prior works.
2. The proposed FC-CLIP simplifies the multiple forwards of the image encoder into just one forward pass, which saves the computation cost.
3. The proposed model outperforms many prior works and more quantitative results are also provided in the supplementary material.

**Weaknesses:**

1. In the related work, I would suggest authors discuss more about the relationship with F-VLM, which also uses the frozen Convolution CLIP as the shared image encoder. What are the major differences between the designs of F-VLM and FC-CLIP?
2. The ablation study on CLIP model type is missing. In Figure 1, the authors demonstrate that CNN-based CLIP should get better features than ViT-based CLIP. I truly appreciate that the authors did compare ViT-L/14 with ConvNeXt-L in Table 5 of the supplementary material. I would suggest authors also compares other CNN-based CLIP like R50x4, R50x16, R50x64 used in F-VLM.
3. Table 5 in supplementary material shows a very interesting experiment, which shows increasing resolution from 224->448, ViT-L still outperforms ConvNeXt-L. But when increasing resolution to 672, ViT-L/14 accuracy drops significantly, which is slightly counter intuitive. Is this PQ evaluated with geometric mean or not? And I would also recommend evaluating zero-shot ImageNet classification accuracy at different resolution inputs, which could better justify whether CNN-based CLIP outperforms ViT-based CLIP when changing resolutions.
4. Authors state "training model solely with ViT-based CLIP is infeasible", probability due to the GPU memory constraint. To compare more fairly with CNN-based CLIP, I would suggest authors use a sliding window to extract features for ViT-based CLIP, e.g. sliding a 224x224 or 336x336 window over 1024x1024 input image to extract the ViT-based CLIP features, which should still have the capability of zero-shot classification.

**Questions:**

1. I noted authors used a different resolution setting, long side 1281. Is there any ablation on this design compared to Mask2Former 1024 short-side resize used in other works? Besides, regarding the inference time comparison, I would suggest authors use the same input size for all the models.
2. In Table 1 of supplementary material, I think the last row should be swapped with the second last row.

---

> ### Author Rebuttal · Authors · 2023-08-09
>
> We sincerely thank the reviewer for the constructive feedback and address the concerns below.
>
> >***W1: Relationship with F-VLM***
>
> Please refer to **C1: Relationship to F-VLM**
>
> >***W2: Other CNN-based CLIP***
>
> We thank the reviewer for the valuable suggestion. We provide results with different CNN backbones below:
>
> | CLIP      | R50 (38.3M) | R101 (56.3M) | R50x4 (87.1M) | R50x16 (167.3M) | ConvNeXt-L (199.8M) |
> |-----------|:-----------:|:-----------:|:------------:|:---------------:|:-------------------:|
> | ADE20K PQ |     15.5    |     17.1    |     18.5     |       20.4      |         24.5        |
>
> As shown in the table above, the performance increases as the model size increases. Using ConvNeXt-L backbone in our setting achieves the best performance, echoing that ConvNeXt is a more modern design of CNN. We note that prior arts use a very strong backbone and CLIP model (e.g., StableDiffusion UNet + ViT-L/14 as in ODISE), which makes it hard to compare against them with much smaller models. Therefore, we adopt ConvNeXt-L in the end (note that ConvNeXt-L has 199.8M parameters, while ViT-L has 304.3M).
>
> >***W3: Table 5 in Supplementary***
>
> We thank the reviewer for the question and suggestion. We address each concern below.
>
> **Evaluation setting**: We herein provide more details regarding the mask classification evaluation. The experiments are our early exploration to verify the effect of using CNN-based and ViT-based CLIP as mask classifiers at different input resolutions, with ODISE’s codebase. Specifically, we adopt ODISE’s mask proposals (thus the mask proposals are kept the same across settings, and ***not trained*** for any specific CLIP model), but replace the mask classification results using either a LAION-2B pretrained ViT-L/14 or ConvNeXt-L CLIP backbone. For ViT-L/14, the classification pipeline follows MaskCLIP [24] and ODISE with attention masking. For ConvNeXt-L, we employ the simple mask pooling to obtain classification logits for each mask. For simplicity and a fair comparison, there is ***no*** geometric mean used in this ablation study.
>
> **Zero-shot ImageNet accuracy**: We appreciate the suggestion and perform a similar experiment on ImageNet in the following table.
>
> | IN1k Acc   |  224 |  336 |  448 |  560 |  672 |  784 |  896 |
> |------------|:----:|:----:|:----:|:----:|:----:|:----:|:----:|
> | ViT-L/14   | 75.3 | 74.3 | 71.3 | 67.5 | 63.1 | 58.5 | 53.9 |
> | ConvNeXt-L | 75.1 | 77.1 | 76.8 | 74.2 | 69.8 | 65.6 | 58.4 |
>
> We observe the same trend (as in segmentation) that while both ViT-based and CNN-based CLIP can yield a strong zero-shot performance at smaller resolutions, CNN-based CLIP generalizes better when scaling up input resolutions.
>
> We would like to emphasize that the scenario of applying the CLIP model for image classification on ImageNet and mask classification on COCO is slightly different. ImageNet contains object-centric images, while COCO has objects with diverse scales. As a result, scaling up input resolution from 224 to 448 improves the recognition accuracy (as small objects are more visible) for **both** ViT-based and CNN-based CLIP models on COCO (see Table 5 in Supplementary), but not on ImageNet.
>
>
> >***W4: ViT-based CLIP backbone in a sliding window manner***
>
> We thank the reviewer for the question and suggestion. We note that using ViT-based CLIP backbone presents several technical challenges, including, but not limited to: GPU memory, absence of multi-scale features, resolution discrepancy between upstream pretraining and downstream segmentation, etc. Even with the suggested sliding window method, a careful re-design is still needed to handle problems such as overlapping pixels, ensemble class tokens at each window, a side-adapter to generate multi-scale features, etc. As noted in the paper, we think a CNN-based CLIP is a straightforward and effective solution to these problems, allowing us to build a simple, strong, and effective single-stage open-vocabulary segmentation framework. That being said, we agree with the reviewer that adapting ViT models is an interesting and promising research problem, especially considering that ViT usually demonstrates a better property of model scaling up.
>
> >***Q1: Resolution difference***
>
> We thank the reviewer for the question. We note that FC-CLIP is a meta architecture that can build upon several different segmentation frameworks. In the submission, we follow kMaX-DeepLab to resize longer edge to 1281 and pad shorter edge to 1281, which gives a similar effective size (if only considering non-padded pixels) as Mask2Former which resizes shorter edge to 800 and longer side to 1333. To further clarify the resolution issue, we also provide results of building FC-CLIP upon Mask2Former in the following table. Note that ODISE is also built upon Mask2Former and thus FC-CLIP-Mask2Former further provides a comprehensive comparison with ODISE.
>
> |                     | frozen params (M) | trainable params (M) |             ADE20K        | Mapillary Vistas | Cityscapes |
> |---------------------|:-----------------:|:--------------------:|:---------------------:|:----------------:|:----------:|
> | ODISE               |        1494       |          28          | 22.6 / 23.4 (caption) |       14.2       |    23.9    |
> | FC-CLIP-kMaX        |        200        |          34          |          24.5         |       17.0       |    43.0    |
> | FC-CLIP-Mask2Former |        200        |          21          |          26.8         |       18.2       |    44.0    |
>
> As shown in the table above, we observe FC-CLIP even achieves better performances, when we switch to the same segmentation framework as ODISE (i.e., use Mask2Former).
>
> >***Q2: Swap rows in Table 1 of sup***
>
> Thanks for the suggestion. We will look into it in the revision.

---

> > ### Comment · Reviewer_N4iz · 2023-08-20
> > **Response to author**
> >
> > Thank the authors for the detailed rebuttal. I like it a lot. I am glad to see Mask2Former further improve the performance of FC-CLIP with less trainable parameters. They are both great frameworks but with some different implementation details. I also truly appreciate authors' effort in open-souce and reproducibility. I would like to raise my rating from Weak Accept to Accept. And please do include more discussions with F-VLM for the general audience.

---

> > > ### Author Response · Authors · 2023-08-20
> > >
> > > Thanks a lot for reading our response and providing the valuable feedbacks! We will incorporate your valuable suggestions into our next revision.

---

### Author Rebuttal · Authors · 2023-08-09

We appreciate all reviewers for their valuable suggestions, and we address the common concerns as follows. For the remaining concerns, please see the individual post for each reviewer.

>***C1: From reviewers N4iz W1, VW2t W1, Relationship to F-VLM***

We thank the reviewers for the suggestion. We note that F-VLM is a pioneering work that builds an open-vocabulary detection framework on top of a frozen CLIP backbone. However, FC-CLIP differs from it with a totally different observation and motivation, as detailed below.

Our work was initially motivated by the state-of-art open-vocabulary segmentation model ODISE, which found that the CLIP backbone extracts noisier features than diffusion models (see Figure B. 1. in ODISE paper), leading to inferior segmentation results (which justifies their adoption of diffusion models). Their observation motivated us to look deeply into the problem. Interestingly, our discoveries show that both ViT-based (used by ODISE) and CNN-based CLIP can produce semantic-meaningful features. However, when scaling up the input resolutions, we discover that ViT-based CLIP features turn noisier, while CNN-based ones are smoother and generalize better across input sizes.

Concurrently, F-VLM also empirically found that a frozen CLIP can provide meaningful features for object detection; however, they did not choose CNN-based CLIP on purpose and thus did not compare carefully between ViT-based and CNN-based CLIP backbones. On the other hand, in our paper, we have provided careful ablation studies on ViT-based and CNN-based CLIP in the Table 5 of Supplementary, where we observe that even though both ViT-based and CNN-based CLIP initially have comparable performance at resolution 224, CNN-based CLIP shows better and more robust performance when input resolution scales up. These important studies are missing in F-VLM.

Finally, even though we would like to provide more technical implementation differences from F-VLM, we note that in their github repository, they only released a demo and testing code (we note that their inference code was just open-sourced on **8/7/2023**, and their best backbone R50x64 is not available), which makes it hard to provide an in-depth comparison and also prevents the community from reproducing their results (please see F-VLM's OpenReview public comment, where people reported failing to reproduce the results). On the contrary, we promise to fully release all the training/testing codes of FC-CLIP (and best backbone) to facilitate the research in this area.

>***C2: From reviewers YCke Q1, VW2t W1, 5KPy W1, Novelty/Contribution***

To be concrete, we summarize our contributions/novelties as follows:

1. To the best of our knowledge, FC-CLIP is the first work that provides an in-depth analysis on adapting different types of CLIP model for downstream open-vocabulary segmentation tasks that require higher resolution inputs, while prior works (e.g., MaskCLIP, ODISE) simply favor a frozen CLIP model or Diffusion Model without looking into the difference between CNN-based and ViT-based CLIP models.

2. We look into the problem of extending the CLIP model for open-vocabulary segmentation, and identify the important problem of resolution discrepancy between the pre-training stage (image-text contrastive learning) and fine-tuning stage (open-vocabulary segmentation), and propose a simple and effective solution by adopting a frozen CNN-based CLIP model.

3. FC-CLIP is a simple and effective meta architecture that can be easily adopted on top of different segmentation frameworks. It not only achieves significantly better performance compared to prior state-of-the-art methods but also enjoys a much lower training/testing cost.

---

### Decision · Program_Chairs · 2023-09-21

**Decision:**

Accept (poster)

**Comment:**

This work proposes an improved method for open-vocabulary panoptic segmentation, which results in significant improvements in accuracy, and training and inference speeds (7x) over the existing SOTA. The key research insight that enabled this is the authors' finding that the image encoder of a ConvNext-based CLIP model scales better to high resolution input images versus a ViT-based backbone, which, in turn, results in better accuracy for segmentation tasks.

The majority of the reviewers were positive (A, WA, WA, BA) except one Reviewer ***YCk*** (R). The latter's major concerns were (a) the lack of empirical comparisons to the existing work OpenSeg [28] and (b) lack of sufficient novelty. The AC and SAC discussed this paper in detail.

OpenSeg [28]'s code is not available and the authors have made an attempt to compare against a close proxy to OpenSeg [28] and shown their method to be better. The paper provides a clear technical novel insight that convolutional backbones from the CLIP encode can help address the issue of scaling up with image resolution, resulting in both significant improvements in accuracy and efficiency over the state-of-the-art for open-vocabulary segmentation. Hence, there is sufficiently novelty and strong experimental results in this work, which justify accepting it for NeurIPS 2023.

The AC and SAC's final decision is to accept this work. Congratulations!

The authors are strongly advised to incorporate the changes in the final manuscript that they promised in the rebuttal.